# The Rare Case of Perirenal Abscess in a Child—Possible Mechanisms and Methods of Treatment: A Case Report and Literature Review

**DOI:** 10.3390/medicina57020154

**Published:** 2021-02-09

**Authors:** Patrycja Sosnowska-Sienkiewicz, Ewa Bućko, Przemysław Mańkowski

**Affiliations:** 1Department of Pediatric Surgery, Traumatology and Urology, Poznan University of Medical Sciences, 60-572 Poznan, Poland; mankowskip@ump.edu.pl; 2Department of Pediatric Surgery, Traumatology and Urology, Karol Jonscher Hospital, 60-572 Poznan, Poland; e.bucko37@gmail.com

**Keywords:** abscess, children, perirenal, renal, surgery

## Abstract

Renal and perirenal abscesses are very rare in children. They can be present as an acute emergency condition or insidiously as a chronic disease. The diagnosis is not so obvious, and it is a big challenge, especially when it can simulate a kidney tumor. The treatment can be conservative, preferably with targeted antibiotics, or surgical, consisting primarily of drainage. This publication aims to present a clinical case in which both diagnosis and treatment were a big challenge for the entire treatment team. A 10-year-old male patient was admitted to the hospital because of mild abdominal pain and a temperature of 37.5 °C. The symptoms lasted for a week. In the computed tomography (CT), the lesion’s dimensions were 11.1 × 8.2 × 25 cm, and inflammation, abscess, cyst, and abdominal tumor have been suggested. The decision about surgical treatment was made. An enormous abscess near the right kidney was localized. The patient’s condition stabilized after surgery. Unfortunately, due to persistent purulent reservoirs, a second laparotomy was necessary. During the extensive diagnostic cystourethrography performed, vesicoureteral reflux was visualized. In conclusion, though a perinephric abscess is very rare in children, it should be taken into consideration in patients with non-specific abdominal symptoms. The imaging using ultrasound and CT scan with contrast enhancement is crucial to recognize and properly treat the condition. In terms of a small abscess, the only antimicrobial treatment using antibiotics of a broad spectrum can be considered. However, the drainage of an abscess, either percutaneous or open, should be used. For the large abscess, the open drainage seems to be a primary method of treatment. The importance of cooperation in a multidisciplinary team is crucial, as the diagnosis and treatment of underlying causes are essential.

## 1. Introduction

Renal and perirenal abscesses are very rare in children [1]. The incidence ranges from 1 to 10 patients for every 10,000 hospital admissions. About one-third of patients with this diagnosis have diabetes. It can be present as an acute emergency condition or insidiously as a chronic disease [2]. The diagnosis is not so obvious, and it is a big challenge, especially when it can simulate a kidney tumor.

The proper diagnostics must be carried out carefully, because improper treatment may result in the loss of the kidney or even death of the patient [3].

Symptoms that may suggest a renal or perirenal abscess include a fever and pain of the abdomen. Therefore, the differential diagnosis of these patients is extremely difficult. For this reason, frequently, this condition is detected as a result of incidental imaging, particularly computed tomography (CT).

An abscess around the kidney may be related to upper tract urosepsis associated with an infective staghorn renal calculus [2,4,5,6]. The second cause may be infection coming from the renal carbuncle or renal abscess perforated into the perirenal space. This cause is more probable in suspected immunocompromised patients. In these instances, Staphylococcus aureus is often diagnosed. Younger or malnourished patients are more likely to have an acute presentation with a high temperature and acute abdomen symptoms.

The treatment can be conservative, preferably with targeted antibiotics, or surgical, consisting primarily of drainage.

This publication aims to present a clinical case in which both diagnosis and treatment were a big challenge for the entire treatment team.

## 2. Case Report

A 10-year-old male patient was transferred to the surgical department from the pediatric ward due to severe symptoms of the abdomen. The child was admitted to the hospital because of mild abdominal pain and a temperature of 37.5 °C. The symptoms lasted for a week.

In the infectious disease ward, COVID-19 was excluded. During hospitalization, the patient developed acute abdominal symptoms. The acute condition of the boy appeared suddenly. Results of laboratory tests showed deviations in white blood cells (WBC) = 11.89 × 10^3^/µL, and hemoglobin (HGB) = 6.9 g/dL, with hematocrit (HCT) = 20.6%. The inflammatory markers were elevated. Urinalysis showed 75 mg/dL of proteins. The ultrasound of the abdomen and computed tomography were performed. The diagnosis was unclear. Inflammation, abscess, cyst, and abdominal tumor were suggested. In the computed tomography (CT), the lesion had dimensions: 11.1 × 8.2 × 25 cm (Figure 1 and Figure 2).

It displaced and compressed the inferior vena cava and right renal vein. The lesion adhered to and modeled the liver and pancreas. It was pushing the intestinal loops. The lesion reached down into the bladder. After multidisciplinary consultation with a radiological and oncological team, the decision about surgical treatment was made. Exploratory laparotomy was performed. During the surgery, there was a large amount of purulent content in the peritoneal cavity. A large abscess was visible in the vesicorectal recess. An enormous abscess near the right kidney was localized. The abdominal cavity was thoroughly rinsed. The abdominal cavity and the perirenal space were drained. The bladder was thick and fibrous.

Antibiotic therapy was started. MSCNS (Staphylococcus coagulase-negative susceptible to methicillin), a strain susceptible to beta-lactam antibiotics except for penicillin, amino-, carbo-, and ureidopenicillins, was grown from the inoculation of the material collected from the abdominal cavity. The urine, blood, and tracheal aspirate cultures were also collected and showed no abnormalities. After surgery, the patient’s condition improved slightly. On the seventh day after the procedure, the patient again presented acute symptoms of the abdomen. He had a fever. The decision about relaparotomy was made. A vast amount of purulent content was evacuated from the vesicorectal and right perirenal recesses one more time. Additionally, the bladder, with large amounts of residual urine, was visualized. The wall of the bladder was thickened.

The peritoneal cavity was rinsed again, and drains were inserted. The patient was catheterized.

A broader spectrum antibiotic was introduced. Due to the significant exhaustion of the patient, gastroenterological consultation was made, and nutritional treatment was introduced. To exclude an accompanying disease responsible for the level of malnutrition, endocrinological consultation was made. Broad laboratory diagnostics excluded any hormonal abnormalities. The patient’s condition was monitored by a CT scan, which showed duplication of the right kidney system and altered parenchyma of the kidney. Periodic control ultrasound of the abdomen showed the progressive reduction of the fluid reservoirs. The patient’s condition improved slightly. A febrile episode of up to 38.5 °C occurred, and infection of the central intravenous catheter was suspected. After its removal, the general condition of the boy was improved. The drains from the abdomen were removed. When an attempt was made to take out the catheter from the bladder, the inflammatory parameters increased after 24 h. There was a fever. The abdomen was tight and tender. After re-catheterization of the patient, 500 mL of residual urine was evacuated. The unwanted symptoms disappeared. The patient’s condition was stabilized.

After the urological consultation, a cystoscopy was performed. It showed a bladder with abnormal trabeculation. The cystofix was fixed. Broad-spectrum antibiotic therapy was maintained. Laboratory inflammatory markers improved gradually. No microbial growth was achieved in the control cultures. The last antibiotics were discontinued after seven days.

A multidisciplinary team consisting of an oncologist, immunologist, and nephrologist decided the necessary diagnostics and treatment. After the extensive physical, laboratory, and radiological studies, the oncological consultation determined that the probability of neoplastic growth was low. Acquired immunodeficiency, such as AIDS or tuberculosis, was excluded by an infectious disease specialist after appropriate laboratory tests. A detailed immunological study was performed. On this basis, functional disorders in the scope of class switching and possible quantitative B lymphocytes were found. The distribution of T-cytotoxic lymphocytes may indicate functional disorders. Abnormalities were diagnosed in the phagoburst tests and the percentage of lymphocytes, and the tests required repeating and control, and suggested an immune disorder.

From the moment of the surgery, the patient’s condition improved and stabilized significantly.

A cystourethrography study was performed and showed the fifth grade of vesicoureteral reflux and duplicated collecting system on the right side (Figure 3).

The patient was transferred to the nephrology clinic for further treatment.

The control ultrasound examination of the abdominal cavity did not reveal the abscess that had been operated on before. After further gastroenterological consultation, the boy started to gain weight. He was discharged after a seven-week hospitalization, with recommendations for further meticulous nephrological and urological control.

The 99mTc-MAG3 diuretic renal scintigraphy was performed in our patient. The normal value of secretory and excretory renal function was obtained in the study. In the future, the patient will undergo surgical treatment.

Despite the detailed diagnostics, no specific cause of the perirenal abscess was found.

## 3. Discussion

A pediatric perirenal abscess diagnosis is challenging due to its infrequent occurrence and non-specific symptoms—fever and flank pain. Because of the non-specific presentation, it can be confused with other abdominal diseases. The differential diagnosis contains retrocaecal appendicitis, Wilms tumor, or neuroblastoma [2]. Moreover, symptoms pointing to renal involvement, such as dysuria or frequent urination, are usually not present, as in this case [5,6,7]. In the presented case, the patient suffered for a week from fever up to 38 °C and abdominal pain. He was also cachectic. The symptoms presented at admission pointed strongly to neoplasm. That is why the use of detailed diagnostics is essential.

Routine blood work, including complete blood count (CBC) and renal parameters, should be performed. Many authors point to leukocytosis and anemia in only half of the patients [8,9,10,11,12]. Urinalysis should be performed to assess pyuria or proteinuria. However, Saiki et al. claim that normal results of urinalysis can be presented in up to 30% of patients when there is restricted communication between an abscess and the urinary tract [12]. Hematuria is present in only 6–30% of cases, despite the fact that the urinary tract pathology is common comorbidity [13]. None of the laboratory results are specific to the perirenal abscess.

After consultation with the nephrological team, our patient underwent a panel of tests assessing kidney function. We should remember the negative effect of asymptomatic hyperuricemia on the kidneys. Therefore, it is worth emphasizing the role of urine sediment analysis in kidney diagnosis. The analyses performed on our patient showed no deviations [14].

As laboratory diagnostics usually cannot deliver a conclusive answer, radiological diagnostics should always be used. The recommended means of radiological diagnostics consist of ultrasonography, followed by CT scan with contrast enhancement [4]. However, the imaging can mimic other conditions, such as Wilms tumor, renal cell carcinoma, or xanthogranulomatous pyelonephritis. In the presented case, the CT scan was chosen as a primary diagnosis method, as a neoplastic tumor was suspected. The study was not conclusive, but it suggested an inflammatory process. Since the final diagnosis could not be reached, a diagnostic laparotomy was made. In the presented case, the diagnosis was made intraoperatively on day three of hospitalization; however, the severity of the illness suggested a much longer process [4]. As the complications range from sepsis and fistula to peritoneal rupture or perforation through a diaphragm [7], a fast diagnosis seems crucial to avoid an increase in morbidity. Therefore, in the case of uncertain imaging combined with a lack of clinical improvement, surgical exploration seems to be the proper solution to diagnostic doubts. The suggested treatment for a perirenal abscess is antibiotic therapy covering both aerobic and anaerobic pathogens and drainage of the abscess [15,16]. In the publication by Coelho et al., interventional treatment for a perinephric abscess was preferable, either by performing percutaneous drainage or open surgical drainage [3]. The publication of Angel et al. acknowledges the possibility of only antibiotic therapy for an abscess less than 3 cm in diameter. However, in the case of failure of antibiotic therapy treatment and the delay of an invasive procedure, the need for nephrectomy may occur. Therefore, to avoid complications, the authors recommend the invasive procedure [16]. In the presented case, antibiotic therapy was introduced on the first day of admission, with no improvement in the patient’s condition. The size of the lesion visualized was 11 × 8 × 25 cm. As was mentioned, due to uncertain diagnosis, laparotomy, washing, and consequent drainage were performed. According to the analyses, the use of catheter drainage (i.e., leaving drainage in the abdominal cavity permanently) is more effective than percutaneous drainage. The extent of the abscess suggests that percutaneous drainage would not be sufficient in that case. There is always a method of drainage to consider, or exploratory laparotomy vs. a direct surgical approach [17,18]. Moreover, despite the extensive irrigation used during the first procedure, after a week, the patient experienced a recurrence of pain and fever. Another intervention should be considered in the case of extensive abdominal abscess and symptoms presented post-operatively after the fourth day of surgery [17]. In this case, the decision for relaparotomy was made. Washing and drainage of the cavity were used again. Drainage, after both the first and second operation, was maintained for seven days, after the first laparotomy until the second intervention. Furthermore, after the second intervention, drainage was maintained for seven days until the amount of fluid in the abdominal cavity was almost completely absorbed. In both cases, it was classic drainage without flushing. After the surgical intervention, the patient’s condition improved.

Gardiner et al. propose that the most common pathophysiology for perirenal abscess is ascending urinary tract infection, often associated with a staghorn calculus. The other causes may come from renal carbuncles or renal abscess perforating the perirenal space. This is probable for immunocompromised and younger patients [3]. There is also a less common hematogenous pathway [12]. In the case of our patient, all of the pathways were taken into consideration.

According to Gardiner et al., the most common Gram-negative pathogen is *Escherichia coli*, and its presence implies the ascending origin of the abscess. For that path, pathogens like *Klebsiella* spp. and *Enterobacter* spp. should also be considered [3]. The presence of Staphylococcus Aureus suggests the hematogenous spread [16].

In the present case, MSCNS was the isolated pathogen. This supports the ascending urinary tract infection hypothesis, because this is a pathogen often observed in urinary tract infection (UTI) [5,8]. Coagulase-negative Staphylococcus is one of the common pathogens causing UTI in children with anatomical or functional abnormalities of the urinary tract or a compromised immune system [9].

In the publication by Rote et al., the authors suggested that, in the case of renal parenchymal infection with Gram-negative bacteria, the patient should undergo cystourethrography [1]. In our clinic, such a procedure was performed, as it was suspected that the origin of the process ascended from the urinary tract. The presence of fifth-grade vesicoureteral reflux and duplication of the ureter, combined with the abnormalities of the bladder, pointed to the increased risk of ascending UTI [6,8].

The underlying cause of the perirenal abscess should be diagnosed and treated. The urinary tract obstruction should be excluded or treated appropriately [2]. In the presented case, cystoscopy was performed. Except from an abnormal bladder, no additional obstruction was visualized. It is suspected that thickened walls and bladder trabeculation were caused by chronic bladder inflammation of uncertain origin. The cystofix was fixed in order to facilitate urinary flow, which significantly improved the patient’s condition.

Another underlying cause of the abscess might be the compromise of the immune system, diabetes mellitus, or coexisting chronic disease [2,4]. Screening must be performed in order to remove accompanying problems. In our case, the only abnormality was present in the phagoburst tests and the percentage of lymphocytes. Both of the tests might suggest immunodeficiency; however, they should be repeated, as the patient’s condition will be more stable.

## 4. Conclusions

In conclusion, though perinephric abscess is very rare in children, it should be taken into consideration in patients with non-specific abdominal symptoms. Diabetes mellitus, urinary tract abnormalities, and immunodeficiency should be considered while treating a patient with perirenal abscess. Laboratory examination is necessary to stabilize the patient, but does not facilitate the diagnosis. Imaging using ultrasound and CT scan with contrast enhancement is crucial to recognize and properly treat the condition. In terms of a small abscess, only antimicrobial treatment using antibiotics of a broad spectrum can be considered. However, in any doubt, the drainage of an abscess, either percutaneous or open, should be used. For a large abscess, open drainage seems to be the primary method of treatment.

The importance of cooperation in a multidisciplinary team is crucial, as the diagnosis and treatment of underlying causes are essential.

## Figures and Tables

**Figure 1 medicina-57-00154-f001:**
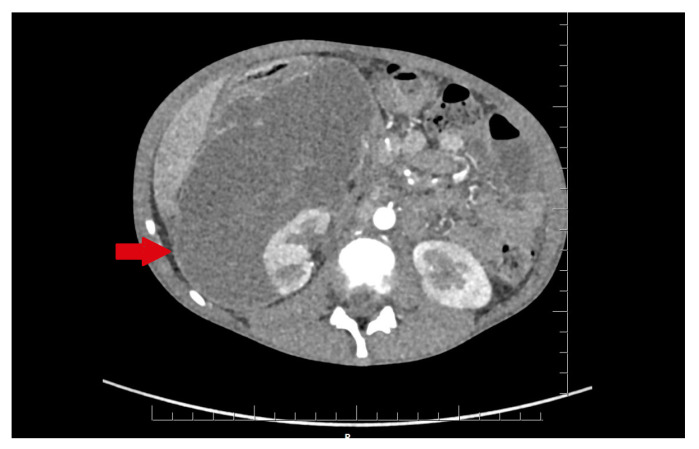
The lesion of unknown character in the area of the right kidney, shown in the transverse plane of the CT scan. The lesion is marked by an arrow.

**Figure 2 medicina-57-00154-f002:**
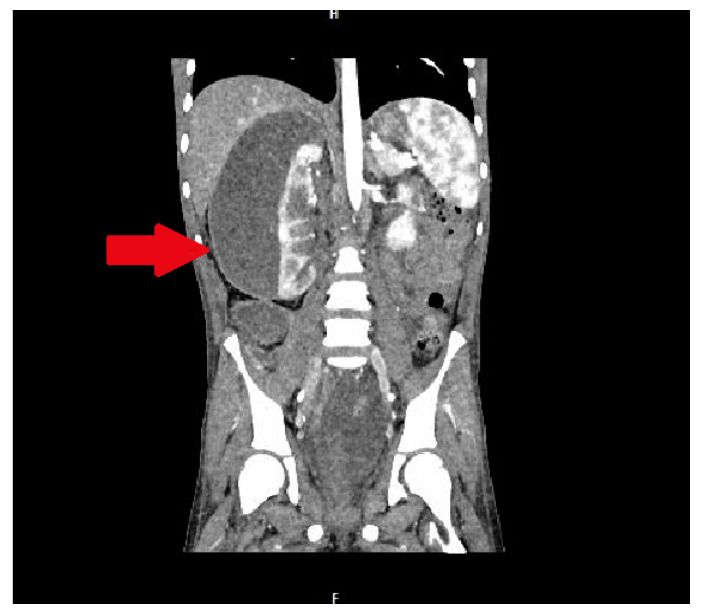
The lesion of unknown character in the area of the right kidney, shown in the frontal plane of the CT scan. The lesion is marked by an arrow.

**Figure 3 medicina-57-00154-f003:**
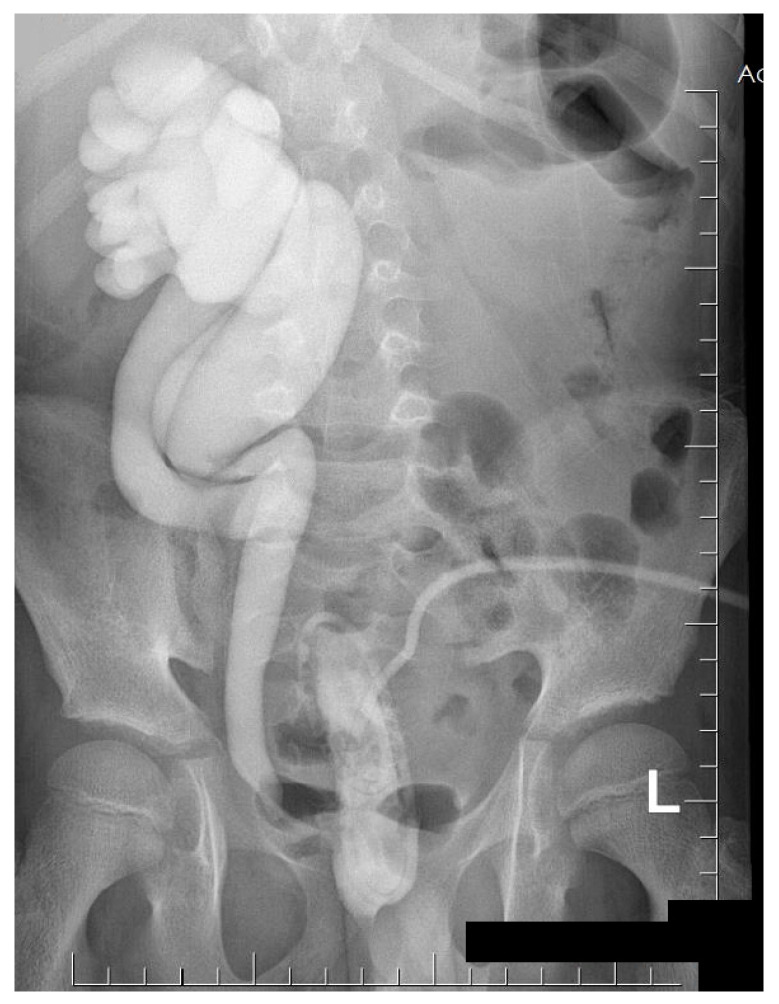
The duplicated collecting system and the fifth-degree vesicoureteral reflux on the right side were presented in the micturating cystourethrography.

## Data Availability

Data available on request due to restrictions.

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
