# Peer review of "The Rare Case of Perirenal Abscess in a Child—Possible Mechanisms and Methods of Treatment: A Case Report and Literature Review"

_medicina, 2021, doi:10.3390/medicina57020154_

Round 1
Reviewer 1 Report
In figure 1-2 please place an arrow to show the position of the abscess
In figure 3 it is appropriate to delete the name of the patient
Laboratory data (leukocyte count, neutrophils etc) might be presented in the case presentation rather than in the discussion .
An Ultrasound image of the affected kidney also might be interesting.
The role of the urine sedimenti might also be discussed in view of the publication PMID: 29689561
Author Response
Dear Reviewer,
I am very grateful for the review of the article “The rare case of perirenal abscess in a child- possible mechanisms and methods of treatment: a case report and literature review.” I would like to address your comments and suggestions.
- In figure 1-2 please place an arrow to show the position of the abscess.
Thank you for your appropriate advice. Of course, I added the arrows pointing to the lesion in both Figures 1 and 2.
- In figure 3 it is appropriate to delete the name of the patient.
Thank you for your advice. It was the name of the hospital. Unnecessary information was removed from the figure.
- Laboratory data (leukocyte count, neutrophils etc) might be presented in the case presentation rather than in the discussion .
Thank you very much for your advice. It is of course very right to present the results of the laboratory tests along with the case presentation. I changed the position of this information and now they are transferred to the part: case report.
Results of laboratory tests showed deviations in WBC 11,89 10^3/µl, HGB 6,9 g/dl with HCT 20,6%. The inflammatory markers were elevated. Urinalysis showed 75 mg/dl proteins. (Page 2 Line 63-65)
- An Ultrasound image of the affected kidney also might be interesting.
I am very grateful for that notice. In this publication, it would be valuable to present the ultrasound results. Unfortunately the examination was performed but the images were not saved in the system. We don’t have access to the study anymore.
- The role of the urine sedimenti might also be discussed in view of the publication PMID: 29689561
Thank you very much for your good advice. This publication will additionally increase the value of the discussion in our article. I included the relevant fragment in the discussion and supplemented the references [14)]. The rest of the citations was renumbered accordingly (Page 5 Line 157-161).
After consultation with the nephrological team, our patient underwent a panel of tests assessing kidney function. We should remember the effect of asymptomatic hyperuricemia on the kidneys. It can have negative effects on them. Therefore, it is worth emphasizing the role of urine sediment analysis in kidney diagnosis [14]. The analyzes performed on our patient showed no deviations.
Once again, thank you for your review and valuable comments.
Yours faithfully,
Patrycja Sosnowska-Sienkiewicz, MD, PhD

Reviewer 2 Report
The paper presented a relatively rare case of paediatric perirenal abscess, focusing on difficulty that may be find in diagnosis, treatment and in understanding the underlying causes.
The article needs a general revision of the form both as regards English language and as regard technical/scientific presentation.
Contribution of each specialist in the management of this difficult case could be better explained.
Laboratory results such as WBC, HGB, HCT… (page 5, lines 144-147) have to be reported in the description of the case. The discussion has to focus to insights.
Image captures are too generic, they do not describe what images depicts (consider to ask a radiologist for better explanation); furthermore, the images presented aren’t compatible with an abdominal tumor but with a subcapsular (not perirenal) fluid collection.
Image 1 and 2 do not depict hydronephrosis: how do you explain the following cystouretrography with important dilation of the urinary tract?
Do you find an explanation for urine retention?
Author Response
Dear Reviewer,
I am very grateful for the review of the article “The rare case of perirenal abscess in a child- possible mechanisms and methods of treatment: a case report and literature review.” I would like to address your comments and suggestions.
- Contribution of each specialist in the management of this difficult case could be better explained.
Thank you for the advice. In such a complicated case, the assessment of the multidisciplinary team is crucial. Our patient was consulted by specialists of:
Radiology
After multidisciplinary consultation with a radiological and oncological team the decision about surgical treatment was made. (page 3 line 77-78)
Oncology
After multidisciplinary consultation with a radiological and oncological team the decision about surgical treatment was made. (page 3 line 77-78)
A multidisciplinary team consisting of an oncologist, immunologist and nephrologist decided about the necessary diagnostics and treatment. After extensive physical, laboratory and radiological studies the oncological consultation determined that the probability of neoplastic growth is low. (page 3 line 115-118)
Gastroenterology
Due to the significant exhaustion of the patient, gastroenterological consultation was made and nutritional treatment was introduced.(page 3 line 96-98)
Endocrinology
To exclude accompanying disease responsible for the level of malnutrition endocrinological consultation was made. Broad laboratory diagnostic excluded any hormonal abnormalities. (page 3 line 98-100)
Urology
After the urological consultation a cystoscopy was performed. It showed a bladder with abnormal trabeculation. (page 3 line 110-111)
Immunology
A multidisciplinary team consisting of an oncologist, immunologist and nephrologist decided about the necessary diagnostics and treatment. (page 3 line 115-116)
A detailed immunological study was performed. On its basis, functional disorders in the scope of class switching and possible quantitative B lymphocytes were found. The distribution of T-cytotoxic lymphocytes may indicate functional disorders. Abnormalities were diagnosed in the phagoburst tests and the percentage of lymphocytes - the tests required repeating, control, and suggested an immune disorder. (page 3 line 119-124)
Nephrology
A multidisciplinary team consisting of an oncologist, immunologist and nephrologist decided about the necessary diagnostics and treatment.(page 3 line 115-116)
The patient was transferred to the nephrology clinic for further treatment. (page 4 line 132)
After consultation with the nephrological team our patient underwent a panel of tests assessing kidney function. (page 5 line 157-158)
Infectious disease
Aquired immunodeficiency such as AIDS or tuberculosis were excluded by infectious disease specialist. (page 3 line 118-119)
This cooperation vastly contributed to diagnosis of the underlying cause and therefore a proper treatment.
Thank you once again for the suggestion to emphasize the role of each member of the team.
- Laboratory results such as WBC, HGB, HCT… (page 5, lines 144-147) have to be reported in the description of the case. The discussion has to focus to insights.
Thank you very much for your advice. It is of course very right to present the results of the laboratory tests along with the case presentation. I changed the position of this information and now they are transferred to the part: case report.
Results of laboratory tests showed deviations in WBC 11,89 10^3/µl, HGB 6,9 g/dl with HCT 20,6%. The inflammatory markers were elevated. Urinalysis showed 75 mg/dl proteins. (Page 2 Line 63-65)
- Image captures are too generic, they do not describe what images depicts (consider to ask a radiologist for better explanation); furthermore, the images presented aren’t compatible with an abdominal tumor but with a subcapsular (not perirenal) fluid collection.
Thank you for your suggestion, it is really helpful. I reconsulted with our radiological team. We precised image captures:
Image capture for Figue 1. The lesion of the undefined character in the area of the right kidney shown in the transverse plane of the CT scan. The lesion is marked by an arrow. (page 2 line 69-70)
Image capture for Figure 2. The lesion of the undefined character in the area of the right kidney shown in the frontal plane of the CT scan. . The lesion is marked by an arrow. (page 3 line 72-73)
Image capture for Figure 3. The duplicated collecting system and the fifth-degree vesicoureteral reflux on the right side were presented in the micturating cystourethrography. (page 4 line 129-130)
In the opinion of our multidisciplinary team including radiologists, there was a suggestion also of abdominal tumor. The differential diagnosis between the tumor, abscess or inflammatory process was difficult for the entire team. It was based on presented images of the CT scan combined with results of ultrasound studies and general patient examination. Precise localization of the fluid was also impossible to assess based on the scans. Despite the multiple consultation with the radiological and oncological team, it was impossible to define the final character of the lesion before surgery.
- Image 1 and 2 do not depict hydronephrosis: how do you explain the following cystourethrography with important dilation of the urinary tract?
Thank you for pointing this problem out. We believe that the images made before surgery did not show the abnormalities because the kidney and ureter were compressed and deformed by the vast fluid collection surrounding the organs. After the surgical fluid evacuation, the pathology became visible.
- Do you find an explanation for urine retention?
Thank you for that question. It is very hard to find a definite explanation for all the abnormalities in such a complicated case. We do believe an abnormal bladder caused the urine retention. The walls of the bladder visualized during cystoscopy were thickened, probably because of the chronic inflammatory process, therefore, causing the dysfunction. As for the pathogenesis of the inflammation, it could be caused either by urinary tract infection, which started prior to hospitalization and was also responsible for creating the abscess. The other reason for the inflamed bladder would also be constant irritation by the purulent fluid in the abdomen, which can also cause the infection.
It is suspected that thickened walls and bladder trabeculation were caused by chronic bladder inflammation of uncertain origin.(page 6 line 226-227)
Once again, thank you for your review and valuable comments.
Yours faithfully,
Patrycja Sosnowska-Sienkiewicz, MD, PhD

Reviewer 3 Report
The present case report focussing on a perirenal abscess in a child thereby reviewing the clinical course. The topic is very interesting for the medical reader, not only working in this field.
Some items should be ameliorated and discussed in the text:
-In the abstract /summary the finding of the vesicoureteral reflux should be mentionned
-Wouldn`t urogenital tuberculosis to be excluded in such a scenario?
-The authors should discuss whether it woud be reasonable to leave the drains (after surgery) for a longer period or rinse the cavity, e.g, every two days?
-Concluded from this case, woudnt it make more sense to make more often a three-phase contrast media CT of the abdomino-urogenital region? This should be discussed.
-The authors mentioned the immunocompromised situation of the child exhibiting perirenal abscesses. Authors should be more concrete naming potential underlying disorders. E.g. did they measure 25-OH vitamin D levels?
-Has Addison´s disease been excluded here ?
-What are the steps for preventing infection in the future in this child?
Author Response
Dear Reviewer,
I am very grateful for the review of the article “The rare case of perirenal abscess in a child- possible mechanisms and methods of treatment: a case report and literature review.” I would like to address your comments and suggestions.
1.In the abstract /summary the finding of the vesicoureteral reflux should be mentionned
Thank you for this helpful note. Vesicoureteral reflux is the important underlying factor in this case. I did correct the abstract.
During the extensive diagnostic cystourethrography was performed which visualized vesicoureteral reflux.(page 1 line 22-23)
2.Wouldn`t urogenital tuberculosis to be excluded in such a scenario?
Thank you for your insight. After the consultation with infectious disease specialist detailed diagnostic was performed and the urogenital tuberculosis was excluded.
Aquired immunodeficiency such as AIDS or tuberculosis were excluded by infectious disease specialist after appropriate laboratory tests. (page 3 line 118-119)
- The authors should discuss whether it woud be reasonable to leave the drains (after surgery) for a longer period or rinse the cavity, e.g, every two days?
Thank you very much for your valuable comment. The method of drainage could influence the treatment of our patient, especially the need for a second laparotomy.
The need for drainage of the abdominal cavity after the intraoperative diagnosis of an extensive abscess in the area of the right kidney in our patient is obvious.
Patrowski et al. in a conducted extensive meta-analysis (Petrowsky H, Demartines N, Rousson V, Clavien PA. Evidence-based value of prophylactic drainage in gastrointestinal surgery: a systematic review and meta-analyzes. Ann Surg. 2004; 240 (6): 1074-1085. Doi: 10.1097 / 01.sla.0000146149.17411.c5) recommend: “Drains should be omitted after hepatic, colonic, or rectal resection with primary anastomosis and appendectomy for any stage of appendicitis, whereas prophylactic drainage remains indicated after esophageal resection and total gastrectomy.” However, this situation concerns planned operations. In our case, drainage was essential.
In the available literature (Surgical Treatment: Evidence-Based and Problem-Oriented. Holzheimer RG, Mannick JA, editors. Munich: Zuckschwerdt; 2001), aspiration only drainage or catheter dreinage is recommended for abdominal abscesses. The use of catheter dreinage is more effective according to the analyzes. We also used this type of drainage.
There is always a method of drainage to consider, exploratory laparotomy vs. "direct surgical approach". In our case, due to doubts as to the initial diagnosis, exploratory laparotomy was performed.
Based on the available literature, no recommendations have been found according to which the abdominal cavity should be rinsed through a maintained drainage tube in the case of abscesses every given period of time. The possibility of using drainage with periodic irrigation in the case of necrotic pancreatitis has been pointed out (Ke L, Li J, Hu P, Wang L, Chen H, Zhu Y. Percutaneous Catheter Drainage in Infected Pancreatitis Necrosis: a Systematic Review. Indian J Surg. 2016; 78 (3): 221-228. Doi: 10.1007 / s12262-016-1495-9).
“Six studies suggested the nursing staff should routinely irrigate the catheters with normal saline solution every 8 h to drain viscous fluid or fluid with particles.”
In our patient, drainage, both after the first and the second operation, was maintained for 7 days, after the first laparotomy until the second intervention. And also after the second intervention for 7 days until the amount of fluid in the abdominal cavity was almost completely absorbed. In both situations, it was a classic drainage, without flushing it.
After conducting a retrospective analysis in our patient and in the available literature, we would still use the classic drainage without flushing, which served its purpose.
Perhaps the abdominal lavage drainage would increase the effectiveness of treatment after the first laparotomy, unfortunately the available literature does not provide any evidence for this in the case of abscesses, and the justification for using drainage and periodic abdominal lavage is very poorly described.
References were supplemented and the relevant text fragment was added to the discussion (page ... line ...)
As it was mentioned, due to uncertain diagnosis, laparotomy, washing and consequent drainage were performed. According to the analyzes, the use of catheter drainage (i.e. leaving drainage in the abdominal cavity permanently) is more effective than percutaneous drainage. The extent of the abscess suggests that the percutaneous drainage would not be sufficient in that case. There is always a method of drainage to consider, exploratory laparotomy vs. "direct surgical approach." [17,18]. Moreover, despite the extensive irrigation used during the first procedure, after a week the patient experienced a recurrence of pain and fever. Another intervention should be considered in case of extensive abdominal abscess and symptoms presented post-operatively after the day 4th of surgery. [17]. In this case, the decision about relaparotomy was made. Washing and drainage of the cavity was used again. Drainage, both after the first and the second operation, was maintained for 7 days, after the first laparotomy until the second intervention. And also after the second intervention for 7 days until the amount of fluid in the abdominal cavity was almost completely absorbed. In both cases it was a classic drainage without flushing. After the surgical intervention, the patient's condition improved. (page 5 line 185-199)
4.Concluded from this case, wouldn't it make more sense to make more often a three-phase contrast media CT of the abdomino-urogenital region? This should be discussed.
Thank you for that note. Our radiological team states that three-phase contrast CT of the abdomino-urogenital region could be more helpful during the diagnostic process, especially in this complicated case. However, it should be taken into consideration that this type of examination increases the dose of radiation. It is even more important if we consider that this also includes the area of radiological protection. While working with children, our team makes every effort to minimize radiation exposure. The presented case was a very rare exception. Tough useful, three-phase contrast media CT should be used only when absolutely necessary.
5.The authors mentioned the immunocompromised situation of the child exhibiting perirenal abscesses. Authors should be more concrete naming potential underlying disorders. E.g. did they measure 25-OH vitamin D levels?
Thank you for your helpful note. In this case the immunocompromised situation was very probable and concerning and, therefore, after consultations of infectious disease specialist, immunologist and oncologist, the broad diagnostic was performed. Diseases such as AIDS, immunoglobulin deficiency, tuberculosis were excluded. The levels of 25-OH vitamin D were normal. The only abnormalities were presented in functional disorders in the scope of class switching and possible quantitative B lymphocytes and the distribution of T-cytotoxic lymphocytes as well as in the phagoburst tests and the percentage of lymphocytes.
A multidisciplinary team consisting of an oncologist, immunologist and nephrologist decided about the necessary diagnostics and treatment. After extensive physical, laboratory and radiological studies the oncological consultation determined that the probability of neoplastic growth is low. Aquired immunodeficiency such as AIDS or tuberculosis were excluded by infectious disease specialist after appropriate laboratory tests. A detailed immunological study was performed. On its basis, functional disorders in the scope of class switching and possible quantitative B lymphocytes were found. The distribution of T-cytotoxic lymphocytes may indicate functional disorders. Abnormalities were diagnosed in the phagoburst tests and the percentage of lymphocytes - the tests required repeating, control, and suggested an immune disorder. (page 3 line 115-124)
- Has Addison´s disease been excluded here ?
Thank you for that insightful question. After the laboratory examination and consultations with endocrinological team Addison disease was excluded.
To exclude accompanying disease responsible for the level of malnutrition endocrinological consultation was made. Broad laboratory diagnostic excluded any hormonal abnormalities. (page 3 line 98-100)
7.What are the steps for preventing infection in the future in this child?
Thank you for that question. The child remains under the urological and nephrological supervision and undergoes regular controls accompanied by laboratory exams and ultrasound control. In the future the vesicoureteral reflux might be treated surgically to prevent the infections.
He was discharged home after a 7-week hospitalization with recommendations for further meticulous nephrological and urological control.
The 99mTc-MAG3 diuretic renal scintigraphy was performed in our patient. The normal value of secretory and excretory renal function were obtained in the study. In the future, the patient will undergo surgical treatment. (page 4 line 134-139)
Once again, thank you for your review and valuable comments.
Yours faithfully,
Patrycja Sosnowska-Sienkiewicz, MD, PhD

Round 2
Reviewer 2 Report
Dear Author,
Thank you for complete response to all suggestion.
I think that now the paper is ready for publication (after minor language check) and really interesting to read.